# Prior PSMA PET-CT Imaging and Hounsfield Unit Impact on Tumor Yield and Success of Molecular Analyses from Bone Biopsies in Metastatic Prostate Cancer

**DOI:** 10.3390/cancers12123756

**Published:** 2020-12-14

**Authors:** Minke Smits, Kamer Ekici, Samhita Pamidimarri Naga, Inge M. van Oort, Michiel J. P. Sedelaar, Jack A. Schalken, James Nagarajah, Tom W. J. Scheenen, Winald R. Gerritsen, Jurgen J. Fütterer, Niven Mehra

**Affiliations:** 1Department of Medical Oncology, Radboud University Medical Center Nijmegen, Geert Grooteplein Zuid 10, 6525 GA Nijmegen, The Netherlands; kamer_ekici@hotmail.com (K.E.); samhita.pamidimarrinaga@radboudumc.nl (S.P.N.); Winald.Gerritsen@radboudumc.nl (W.R.G.); Niven.Mehra@radboudumc.nl (N.M.); 2Department of Urology, Radboud University Medical Center Nijmegen, Geert Grooteplein Zuid 10, 6525 GA Nijmegen, The Netherlands; inge.vanoort@radboudumc.nl (I.M.v.O.); michiel.sedelaar@radboudumc.nl (M.J.P.S.); jack.schalken@radboudumc.nl (J.A.S.); 3Department of Radiology and Nuclear Medicine, Radboud University Medical Center Nijmegen, Geert Grooteplein Zuid 10, 6525 GA Nijmegen, The Netherlands; James.Nagarajah@radboudumc.nl (J.N.); tom.scheenen@radboudumc.nl (T.W.J.S.); jurgen.futterer@radboudumc.nl (J.J.F.); 4Department of Nuclear Medicine, Technical University, Arcisstraße 21, 80333 Munich, Germany

**Keywords:** CRPC, PSMA-PET, bone biopsy, whole-genome sequencing

## Abstract

**Simple Summary:**

Prostate cancer is currently the fifth leading cause of death in men worldwide. To personalize and guide treatment in prostate cancer, identification of druggable genomic alterations is of major importance. Prostate cancer often metastasizes solely or predominantly to the bones, with molecular analyses on bone biopsies challenging due to technical difficulties to identify and obtain biopsies from high tumor cell containing locations. In our retrospective analysis, we showed a significantly higher success rate in patients where biopsy location was selected by a prior PSMA PET-CT compared to solely CT or MRI. CT-guided biopsies in locations with low Hounsfield units (HUs) and deviation of HUs were associated with a higher proportion of successful histological and molecular biopsies. Based on these results, we designed a simple prediction model for daily clinical practice to increase the success rate of bone biopsies for molecular analyses in prostate cancer to guide precision medicine.

**Abstract:**

Developing and optimizing targeted therapies in metastatic castration-resistant prostate cancer (mCRPC) necessitates molecular characterization. Obtaining sufficient tumor material for molecular characterization has been challenging. We aimed to identify clinical and imaging variables of imaging-guided bone biopsies in metastatic prostate cancer patients that associate with tumor yield and success in obtaining molecular results, and to design a predictive model: Clinical and imaging data were collected retrospectively from patients with prostate cancer who underwent a bone biopsy for histological and molecular characterization. Clinical characteristics, imaging modalities and imaging variables, were associated with successful biopsy results. In our study, we included a total of 110 bone biopsies. Histological conformation was possible in 84 of all biopsies, of which, in 73 of the 84, successful molecular characterization was performed. Prior use of PSMA PET-CT resulted in higher success rates in histological and molecular successful biopsies compared to CT or MRI. Evaluation of spine biopsies showed more often successful results compared to other locations for both histological and molecular biopsies (*p* = 0.027 and *p* = 0.012, respectively). Low Hounsfield units (HUs) and deviation (Dev), taken at CT-guidance, were associated with histological successful biopsies (*p* = 0.025 and *p* = 0.023, respectively) and with molecular successful biopsies (*p* = 0.010 and *p* = 0.006, respectively). A prediction tool combining low HUs and low Dev resulted in significantly more successful biopsies, histological and molecular (*p* = 0.023 and *p* = 0.007, respectively). Based on these results, we concluded that site selection for metastatic tissue biopsies with prior PSMA PET-CT imaging improves the chance of a successful biopsy. Further optimization can be achieved at CT-guidance, by selection of low HU and low Dev lesions. A prediction tool is provided to increase the success rate of bone biopsies in mCRPC patients, which can easily be implemented in daily practice.

## 1. Introduction

Prostate cancer is currently the most common cancer in developed countries and the fifth leading cause of death in men worldwide [1]. Despite optimal initial treatment of the primary prostate cancer, men still will develop metastatic prostate cancer [2,3], currently an incurable disease. Following resistance to androgen-deprivation therapy, metastatic castration-resistant prostate cancer (mCRPC) develops, a heterogeneous disease state showing substantial inter-individual genomic diversity [4,5]. Validated molecular biomarkers to help personalize and guide treatment selection are therefore of major importance [6]. Determination of druggable aberrations and pathways in metastatic prostate cancer include DNA repair, e.g., genes involved in DNA damage sensing, homologous recombination, mismatch repair, as well as the PI3K pathway. Molecular characterization of mCRPC moreover contributes to the understanding of treatment resistance and includes assessment of androgen receptor (AR) splice variants, AR structural variants and mutations. Currently, tissue-based techniques such as immunohistochemistry (IHC), RNA in situ hybridization (RNAish), and next-generation sequencing (NGS) are current tools to personalize and optimize treatment for patients with mCRPC.

To individualize patient care through profiling of a fresh prostate cancer metastasis, the first step is to obtain sufficient and high-quality tumor material for IHC and molecular studies. Prostate cancer often metastasizes to the bones solely (43%) or predominantly (73%) [7,8]. Although a biopsy from soft tissue (nodal and visceral metastases) often provides a sufficient tumor yield [9], obtaining enough tumor cells from a bone biopsy proves more challenging, in part due to technical difficulties regarding biopsy procedure from sclerotic bone metastases, and in identifying bone lesions containing predominant cancerous tissue [10,11]. This is one of the reasons why bone metastatic prostate cancer has been underrepresented in most genomic landscape manuscripts [12,13,14] and underrepresented in biomarker-selected clinical trials, mandating fresh tissue biopsies. Previous studies indicated success rates, defined as any tumor cells found, between 25.5% and 85.7%, and have aimed to assess factors influencing tumor yield from bone biopsies [9,15,16,17,18,19,20]. These variables include level of prostate specific antigen (PSA), lactate dehydrogenase (LDH), and Hounsfield units derived from pre-biopsy computed tomography (CT).

Optimal bone-biopsy site selection is deterministic of a successful outcome. The osseous site of choice is commonly determined by visual inspection of pre-biopsy CT, bone scintigraphy, MRI scan and since recently the prostate-specific membrane antigen (PSMA) positron-emission tomography (PET). In our institute, patients referred for a biopsy are discussed in a weekly multidisciplinary meeting, where a radiologist and/or nuclear physician determines the optimal biopsy site. We observed that since the introduction of PSMA PET-CT imaging, rates of successful bone biopsies increased, with higher yields of adequate tissue.

In a recently published multicenter study [21], a high success rate of 70% for molecular analyses of Gallium-68 (^68^Ga) PSMA PET selected bone lesions in metastatic prostate cancer patients was showed. The standardized uptake value (SUV) was found to be a predictive variable. One of the limitations of this study was that no comparisons were made with a group of patients that received a bone biopsy pre-biopsy selected by alternative imaging modalities.

This retrospective study was performed to compare the success rate of ^68^Ga-PSMA PET-CT for biopsy site selection, in comparison to other imaging modalities, and to identify additional predictors that may strongly associate with tumor yield in bone biopsies. This included additional pre-biopsy and biopsy variables, including Hounsfield and deviation, extracted from the CT-guidance biopsy. Based on these data, we aimed to design a simple prediction model that could be used in daily clinical practice.

## 2. Results

### 2.1. Study Population

A total of 99 patients with a total number of 114 biopsies were considered for selection in this retrospective study. After exclusion, a total number of 110 biopsies from 96 patients were included for this study (Figure 1). In total, 29 out of 62 bone biopsies with a prior PSMA PET-CT scan were also included in our previous publication [21]. Baseline patient characteristics are summarized in Table 1. The overall success percentage for histological confirmation of prostate cancer in the biopsies was 76.4% (84 out of 110 biopsies). Successful molecular characterization was performed in 66.4% of all biopsies and in 86.9% of biopsies with histological confirmation of prostate cancer. There were no procedural complications.

### 2.2. Clinical Parameters

The mean age at biopsy was 67 years (interquartile range (IQR): 48–82). Seven men had hormone sensitive prostate cancer at the time of biopsy. No statistically significant differences were found between the groups in age, hormone status, Gleason score (GS) and prior radiotherapy (Table 1).

PSA and albumin levels differed significantly between patients with histologically successful and negative biopsies (*p* = 0.029 and *p* = 0.040, respectively, Table 1). A non-significantly higher PSA was observed for those patients with a successful molecular biopsy outcome (95.5 vs. 54.0; *p* = 0.126). Other laboratory values were comparable between groups (Table 1).

### 2.3. Imaging and Procedural Characteristics

Prior imaging with PSMA PET-CT appeared useful to select site of bone biopsied, as this resulted in a higher proportion of success (85.5% and 72.6% for histological confirmation and successful molecular characterization, respectively) compared to other imaging (63.9% and 58.3% for CT, and 66.7% and 58.3% for MRI, respectively) (Table 2). Biopsies from the spine resulted in a significantly higher proportion of successful biopsies (histological and molecular) compared to other locations: 95.8% and 91.7% for spine vs. 72.2% and 59.5% for pelvis and 57.1% and 57.1% for other locations. With regard to features derived from the CT scan performed at the biopsy procedure, both lower HU and deviation (Dev) resulted in more successful biopsies (histological and molecular; *p* = 0.025 and *p* = 0.010, respectively for HU; *p* = 0.023 and *p* = 0.006, respectively for Dev; Table 2). Other imaging and procedural characteristics were not significantly different (Table 2). Due to a high proportion of missing data of needle gauge used (54% missing) and the number of cores taken during biopsy (17% missing), no robust analyses could be performed with these variables.

### 2.4. Uni- and Multivariable Analyses

Table 3 summarizes the results of the univariate analysis. Patients with prior PSMA PET-CT were more than three times more likely to have a successful histological biopsy compared to patients with prior CT scan (odds ratio (OR) 3.33). Biopsies from the spine were also more likely to have a successful histological and successful molecular biopsy compared to biopsies from the pelvis (OR 8.88 and OR 7.49, respectively). Both lower HU and Dev were associated with a successful biopsy (histological and molecular) (Table 3). On multivariate analysis Dev alone was significantly associated with biopsy result (OR 0.990, *p* = 0.017 and OR 0.989, *p* = 0.008 for histological and molecular positivity, respectively).

### 2.5. Imaging Prediction Model

Low HU represents osteoblastic and high HU represents a more osteosclerotic, commonly non-tumor-containing lesion, while low Dev represents a more homogeneous lesion and high Dev a more heterogeneous lesion with regard to HU. In Figure 2, an illustration shows two of these different types of bone biopsy lesions, per HU and dev, as imaged on a pre-biopsy CT scan. When we categorized HU and Dev into quartiles, the fourth quartile of both HU and Dev was associated with most negative biopsies. Three groups were defined in an exploratory model: (1) patients with HU and Dev both lower than the 75th percentile, (2) patients with HU or Dev greater than the 75th percentile, and (3) patients with both HU and Dev greater than the 75th percentile. The lowest success rate, as described in Table 4, was seen in lesions with both higher HU and Dev, resembling sclerotic lesions (Figure 2A). There was a statistically significant higher success rate in lesions with both lower HU and Dev, corresponding with sclerotic lesions (Figure 2B). Biopsies from the first group resulted in significant more histological confirmed biopsies compared to those in group 2 and 3: 81.8% vs. 55.6% and 44.4%, respectively (*p* = 0.023). Low HU and Dev (group 1) also resulted in more successful molecular analysis: 77.3% vs. 44.4% and 33.3% (*p* = 0.007) (Table 4).

When the cut-off was set at median for HU and Dev (492.30 and 127.20, respectively), groups with both HU and Dev above median also had significantly fewer successful biopsies.

### 2.6. Druggable Pathogenic Mutations within a Bone-Predominant Cohort

Figure 3 illustrates a summary of the targetable genetic mutations within our cohort. In 14% of our patients, a mutation in the HR-related pathway was found, which are druggable by PARP inhibitors and/or platinum chemotherapy [22,23]. In 3%, we identified mutations in the MMRd pathway resulting in mismatch repair deficiency, druggable with by anti-PD1 checkpoint inhibitors [24]. Finally, 44% of our patients had an activated PI3K pathway which could be treated by PIK3CA or PIK3CB inhibitors, or with AKT inhibitors [25].

## 3. Discussion

Previous studies have shown successful histological results in 25.5–85.7% of biopsies [9,15,16,17,18,19,20]. Tissue with a sufficient amount of tumor cells containing high quality nucleic acids is necessary for further molecular testing. In only a few studies, molecular analyses were also performed on bone biopsies with a success rate of 39–81.7% by whole-exome sequencing or targeted NGS [15,19,20]. In comparison, this study had a diagnostic yield of 76.4% and a sufficient tumor cell percentage to allow for molecular analysis in 86.9% of those biopsies. Table 5 provides a literature overview of previous published studies.

In our study, we investigated the impact of different imaging modalities for biopsy site planning and outcome results. We show that the introduction and utilization of PSMA PET imaging resulted in a higher diagnostic yield, as well as an improved success rates of successful molecular analyses by approximately 15% compared to CT and MRI (Table 2). Previous published radiomic studies utilizing CT-imaging variables show that biopsy success is associated with lesions that are either predominantly radiolucent or have a low mean HU, resulting in higher tumor percentages [15,16,20]. In the current study, we found that homogeneous lesions with low Hounsfield units on CT-imaging contained the highest diagnostic yield and proportion of tumor-containing osseous lesions where molecular profiling by NGS could be performed. We developed a simple prediction model with HU and Dev, where the lowest three quartiles associate with highest diagnostic yield and rates of molecular test success. As shown in this study, a homogeneous (low Dev) hypodense lesion (low HU) is associated with better biopsy results. Implementation of HU and Dev measurements could be prospectively used during CT-guided biopsy to select optimal lesions, preferably from a PSMA-avid lesion. Further utilization of PSMA PET parameters, such as a minimum standardized uptake value (SUV) of PSMA in the region of interest, could further enhance the proportion of successful biopsies. Another retrospective multicenter study indicated that both HU_mean_ and SUV_max_ variables from CT and ^68^Ga-PSMA PET imaging, associate with an outcome of at least 30% tumor content in bone biopsies [21]. In a previous published prospective study of ten mCRPC patients, advanced planning with ^68^Ga-PSMA PET and diffusion-weighted MRI increased diagnostic yield up to 90% on cone-beam CT-guided biopsy [26]. Further prospective studies are needed to assess and validate radiomic signatures to predict bone biopsy outcome utilizing CT, MRI and PSMA PET imaging modalities.

New treatment modalities of mCRPC are based on molecular characterization. The pivotal phase III trial of olaparib in molecular selected patients with aberrations in genes directly or indirectly associated with homologous recombination deficiency, indicated that patients with BRCA1, BRCA2 and ATM derive benefit from targeted therapy with olaparib [27]. In almost 30% of patients that develop mCRPC, aberrations in DNA damage repair (DDR) genes are identified [28]. In addition, screening for patients with immunogenic prostate cancer, associated with mutations in mismatch repair [14,29,30] and CDK12 [31], is advocated within routine care and for clinical trial participation. Other targets include aberrations that activate the PI3 kinase pathway (PI3K) [25]. We identified genetic aberrations in DDRd pathways of DNA sensing, homologous recombination and MMR, as well as recurrent aberrations in PI3K in the bone biopsies in 14, 3 and 44%, respectively (Figure 3).

Particularly for ATM, CDK12 and aberrations in PI3K, it is known that these accumulate following castrate-resistance [32,33].

Our study has limitations due to its unplanned retrospective nature, and multiple bias could be introduced. First, a time bias could influence results, as the oldest biopsies were performed with older imaging modalities, and may affect biopsy results. Biopsies were not performed by the same radiologist, with varying experience in interventional radiology, and differences in equipment. Further, the quantitative attenuations of the lesions were determined retrospectively, but in a blinded manner. We aimed to avoid an intra-observer bias by determining the attenuation by one radiologist in our institute. Biopsy sites were pre-selected in a multidisciplinary meeting, however, at the time of intervention, the pre-selected or alternative lesion was biopsied according to the performing intervention radiologists’ judgement of feasibility and safety. Further, although the needle location in the tumor could be assessed, the exact location of each core sample could not be assessed retrospectively. In addition, missing data, e.g., on needle gauge and number of cores, limited analyses for these factors. We did not include metastasize size as a variable since the Prostate Cancer Working Group has classified the size of osseous lesions as non-measurable by MRI, CT or bone scintigraphy [34]. However, for future studies, solely focusing on PSMA PET-guided biopsies, tumor size should be included. Finally, prior or current use of bone protective agents may influence biopsy outcome, but was not assessed in our study. As the data were from patients only from one academic center, the external validity of our prediction model will have to be validated in other centers including community hospitals.

## 4. Materials and Methods

### 4.1. Study Population

For this retrospective study, the patients with metastatic prostate cancer that were considered were also included in the Center for Personalized Cancer Treatment (CPCT)-02 trial (NCT01855477). Eligibility criteria for this study were patients with metastatic prostate cancer that underwent a biopsy of a bone metastasis in our institute between September 2016 and June 2019. We excluded patients with more than 90 days between prior imaging and bone biopsy. In this study, informed consent was obtained within the CPCT-02 trial from all patients and additional approval was provided by the ethical committee at the University of the Radboud Medical Center Nijmegen (2019-5362, 15 April 2019). All clinical and imaging data were collected retrospectively from the electronic patient records in an electronic case report form (Castor).

### 4.2. Variable Definition

Clinical and imaging variables were pre-defined and collected retrospectively from the electronic patient records. Clinical variables included (1) age at time of biopsy, (2) Gleason score (GS) of primary diagnosis, (3) hormone status at the time of biopsy, (4) prior radiotherapy on biopsied metastases and (5) laboratory values collected up to three weeks before or one week after biopsy. Imaging and procedural variables included (1) prior imaging type (^68^Ga-PSMA PET-CT or 18F-PSMA PET-CT imaging, magnetic resonance imaging (MRI), technetium-99 bone scintigraphy and CT-scan), (2) biopsy location, (3) needle gauge, (4) imaging characteristics during biopsy (Hounsfield units (HU), deviation (Dev) defined as the variation of the HU, region of interest (ROI)) and (5) type of image guidance. One experienced radiologist, blinded to the results, retrospectively determined quantitative attenuations (HU, Dev, ROI) of CT-guided biopsied metastases. When more types of imaging were performed prior to biopsy, the leading imaging was determined from the biopsy report.

### 4.3. Sample Collection, DNA Extraction and Molecular Analysis

Histological and molecular analyses of the bone biopsy were performed within the CPCT-02 trial. When more cores were available from the same biopsy, additional targeted NGS results analyzed in our institute were also used for further analyses. As described by Priestley et al. [35], biopsy cores analyzed within the CPCT-02 trial were first examined by an experienced pathologist for estimation of tumor cellularity on a 6 µm hematoxylin and eosin (H&E) stained section. When the tumor cellularity was estimated >30%, 25 sections of 20 µm were collected in a tube for DNA isolation. Frozen tissues were pulverized in RNAse free MQ (110 µL) using the Qiagen TissueLyzer II (2 min and 25 h) and a steel bead containing Sarsted epp. Genomic DNA was isolated from 50 µL pulverized biopsy with the QIAsymphony DSP DNA Mini kit standard protocol for tissue (50 µL eluate). A total of 50–200 µg of DNA was used for whole-genome sequencing.

If available, cores from the same biopsy were analyzed in our institute. Tissue was fixed in formaldehyde 4% and went through a decalcification process with EDTA. Genomic DNA was isolated from tissue sections (generally 6 × 10 µm) using 5% Chelex-100 and 400 mg proteinase K followed by purification using NaAc and EtOH precipitation. DNA concentrations were measured using the Qubit Broad Range kit (Thermo Fisher Scientific, Waltham, MA, USA). In total, 60 ng DNA was used as input for the library preparation using the TruSight Oncology (TSO500) library preparation kit, as described previously [36]. Libraries were sequenced on a NextSeq 500 (Illumina, San Diego, CA, USA).

### 4.4. Outcomes

The objective of our study was to associate the biopsy result with the type of imaging used to select for the site of the bone metastatic biopsy, and pre-defined laboratory, clinical and imaging variables. First, a biopsy was considered successful if presence of prostate cancer cells could be histologically confirmed. More stringently, we assessed the rate of biopsies with sufficient tumor yield allowing molecular characterization on bone metastatic lesions (≥30% tumor cells on ≥5 mm^2^).

### 4.5. Statistical Analysis

Descriptive statistics were performed for baseline clinical and imaging data. The median and interquartile ranges (IQRs) are reported. Clinical and imaging differences between the groups (successful or negative biopsy) were analyzed using a Chi-Squared test for nominal and categorial variables and Mann–Whitney U-test for continuous variables. Optimal dichotomization per quartile of continuous variables from imaging (HU, Dev, ROI) was established from visual inspection of generated histogram for patients with successful and negative biopsy results. All variables were analyzed using univariable logistic regression analysis with biopsy positivity (histological and molecular) as the dependent variables. Statistically significant variables were tested in a multivariable logistic regression model. A *p*-value < 0.05 was considered statistically significant. All statistical analyses were performed using IBM SPSS statistics version 25.

## 5. Conclusions

With our study, we were able to identify clinical and imaging factors influencing tumor yield in bone biopsies in patients with metastatic prostate cancer. First, a prior PSMA PET-CT improves biopsy outcome with regard to commonly used imaging methods. Second, a prediction tool of quantitative imaging attenuation improved the success rate of bone biopsies by selection of lesions with low HUs and deviation. To validate these findings, further prospective validation is needed.

## Figures and Tables

**Figure 1 cancers-12-03756-f001:**
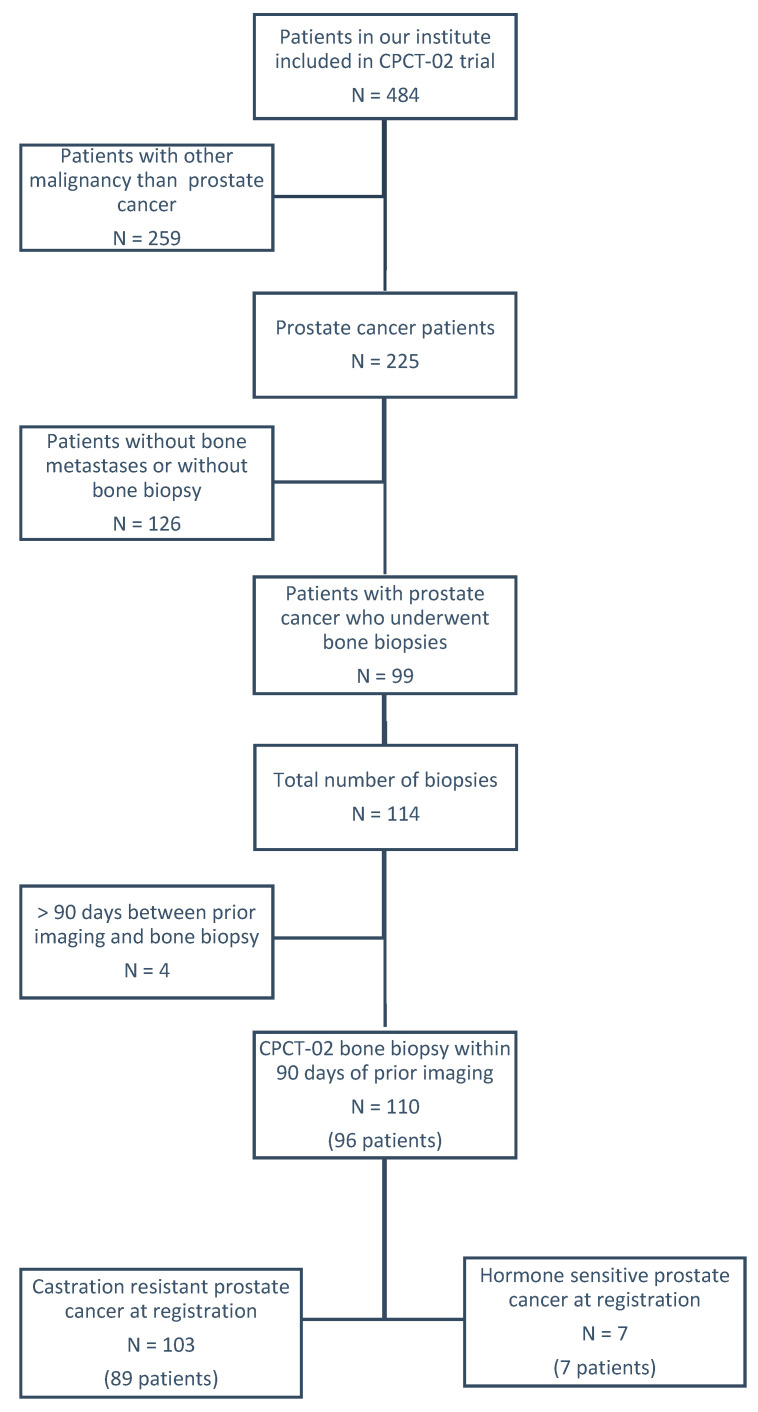
Flowchart.

**Figure 2 cancers-12-03756-f002:**
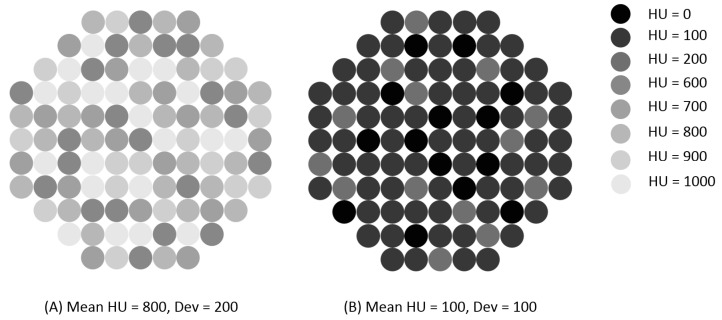
Schematic pictures showing the mean Hounsfield units (HUs) and deviation (Dev) of a lesion on CT scan. (**A**) demonstrates a lesion with a high HU and high Dev. (**B**) demonstrates a lesion with low HU and low Dev.

**Figure 3 cancers-12-03756-f003:**
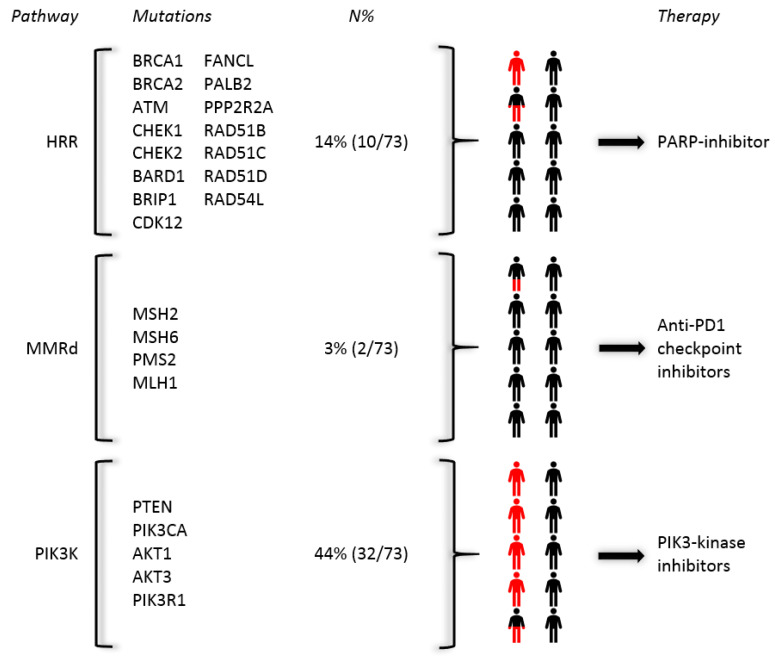
Targetable mutations in our cohort.

**Table 1 cancers-12-03756-t001:** Clinical characteristics and denominators of total biopsies.

Clinical Characteristics	No Tumor Cells Detected	Tumor Cells Present Median (q1–q3) or Percent	*p*-Value *	Insufficient Tumor Yield (<30%)	Sufficient Tumor Yield for Molecular Analysis (≥30%) Median (q1–q3) or Percent	*p*-Value *
Total		76.4% (84/110)			66.4% (73/110) 86.9% (73/84)	
Age at the time of biopsy (years)	65.0 (58.0–72.3)	68.0 (62.0–73.0)	*p* = 0.281	65.0 (59.5–71.0)	69.0 (62.0–74.0)	*p* = 0.137
Hormone status at the time of biopsy			*p* = 0.751			*p* = 0.769
HSPC ^†^	28.6% (*N* = 2)	71.4% (*N* = 5)		28.6% (*N* = 2)	71.4% (*N* = 5)	
CRPC ^‡^	23.3% (*N* = 24)	76.7% (*N* = 79)		34.0% (*N* = 35)	66.0% (*N* = 68)	
Prior radiotherapy on biopsied metastasis			*p* = 0.751			*p* = 0.769
Yes	28.6% (*N* = 2)	71.4% (*N* = 5)		28.6% (*N* = 2)	71.4% (*N* = 5)	
No	23.3% (*N* = 24)	76.7% (*N* = 79)		34.0% (*N* = 35)	66.0% (*N* = 68)	
Gleason score at primary diagnosis			*p* = 0.948			*p* = 0.437
<8		22/28 (78.6%)			18/28 (64.3%)	
≥8		57/72 (79.2%)			52/72 (72.2%)	
Laboratory values						
PSA (µg/L)	38.0 (9.4–110.0)	98.0 (20.5–245.0)	*p* = 0.029	54.0 (12.0–147.5)	95.5 (21.3–315.0)	*p* = 0.126
Alkaline phosphatase (U/L)	109.0 (78.5–140.3)	117.0 (87.0–224.5)	*p* = 0.376	105.0 (78.0–138.0)	128.0 (87.5–232.8)	*p* = 0.100
Albumin (g/L)	34.0 (33.0–36.0)	36.0 (34.0–38.3)	*p* = 0.040	34.0 (33.0–39.0)	36.0 (34.0–38.0)	*p* = 0.567
LDH (U/L)	221.0 (197.8–264.3)	220.0 (180.0–249.0)	*p* = 0.636	220.5 (201.5–264.8)	220.0 (179.0–247.0)	*p* = 0.768
Hemoglobin (mmol/L0)	7.9 (7.4–8.3)	7.7 (6.8–8.3)	*p* = 0.376	7.9 (7.1–8.4)	7.7 (6.9–8.3)	*p* = 0.795
Leukocytes (×109/L)	6.6 (5.5–7.9)	6.1 (4.8–7.9)	*p* = 0.340	6.0 (5.2–7.4)	6.1 (4.8–8.2)	*p* = 0.910
Thrombocytes (×109/L)	236.0 (202.5–287.5)	234.0 (171.3–284.0)	*p* = 0.550	228.5 (180.5–286.3)	238.0 (184.0–286.0)	*p* = 0.448

* Chi-Squared test for nominal variables and Mann–Whitney U-test for continuous variable; ^†^ Hormone sensitive prostate cancer; ^‡^ Castration-resistant prostate cancer.

**Table 2 cancers-12-03756-t002:** Imaging and procedural characteristics.

Imaging and Procedural Characteristics	Tumor Cells Present (*N* = 84)		*p*-Value *	Sufficient Tumor Yield for Molecular Analysis (≥30%)		*p*-Value *
				(*N* = 73)		
	Median (q1–q3) or Number (Percent)			Median (q1–q3) or Number (Percent)		
**Imaging characteristics**						
Imaging type			*p* = 0.037			*p* = 0.292
CT	23/36 (63.9%)			21/36 (58.3%)		
MRI	8/12 (66.7%)			7/12 (58.3%)		
PSMA PET-CT ^¥^	53/62 (85.5%)			45/62 (72.6%)		
(68Ga-PSMA *N* = 52; F18-PSMA *N* = 10)						
Biopsy location			*p* = 0.027			*p* = 0.012
Pelvis	57/79 (72.2%)			47/79 (59.5%)		
Spine	23/24 (95.8%)			22/24 (91.7%)		
Other (3 rib,3 extremity, 1 scapula)	4/7 (57.1%)			4/7 (57.1%)		
**Procedural characteristics**						
Radiologist/fellow Radiologist	56/73 (76.7%)		*p* = 0.942	51/73 (69.9%)		*p* = 0.268
Fellow	17/23 (73.9%)			12/23 (52.2%)		
Internist	11/14 (78.6%)			10/14 (71.4%)		
**Quantitative attenuation**	**No tumor cells**	**Tumor cells present**	***p*-Value**	**Insufficient tumor yield**	**Sufficient tumor yield**	***p*-Value**
HU ^†^	597.6 (327.4–824.9)	447.4 (206.4–579.1)	*p* = 0.025	581.9 (333.0–807.2)	445.8 (182.7–553.1)	*p* = 0.010
Dev ^‡^	174.4 (102.3–223.8)	119.3 (72.0–154.7)	*p* = 0.023	174.6 (104.4–220.7)	111.3 (71.6–147.1)	*p* = 0.006
ROI ^±^	39.6 (33.6–45.0)	34.0 (26.5–41.6)	*p* = 0.108	38.6 (32.3–44.7)	34.0 (26.0–41.8)	*p* = 0.193

* Chi-Squared test for nominal variables and Mann–Whitney U-test for continuous variables; ^¥^ Prostate-specific membrane antigen; ^†^ Hounsfield units; ^‡^ Deviation of lesion; ± Region of interest.

**Table 3 cancers-12-03756-t003:** Univariate logistic regression.

Variable	Successful Histology OR ^†^ (95% CI)	*p*-Value	Successful Genetic Analysis OR ^†^ (95% CI)	*p*-Value
Imaging Type				
CT	A		A	
MRI	1.13 (0.29–4.49)	*p* = 0.862	1.00 (0.27–3.76)	*p* = 1.000
PSMA PET-CT	3.33 (1.25–8.88)	*p* = 0.016	1.89 (0.795–4.496)	*p* = 0.150
Biopsy location				
Pelvis	B		B	
Spine	8.88 (1.13–69.77)	*p* = 0.038	7.49 (1.65–34.09)	*p* = 0.009
Other	0.52 (0.11–2.49)	*p* = 0.409	0.91 (0.19–4.33)	*p* = 0.903
HU	0.998 (0.996–1.000)	*p* = 0.034	0.998 (0.996–1.000)	*p* = 0.016
Dev	0.990 (0.983–0.998)	*p* = 0.017	0.989 (0.981–0.997)	*p* = 0.008
ROI	0.986 (0.953–1.020)	*p* = 0.420	0.992 (0.960–1.025)	*p* = 0.639
ROI log 10	0.110 (0.003–3.797)	*p* = 0.222	0.186 (0.006–5.365)	*p* = 0.327

^†^ odds ratio, A: The CT was used as a reference to compare the other variables (MRI and PSMA PET-CT) to, as also stated in the text above the table; B: The pelvis was used as a reference to compare the other variables (Spine and other) to, as also stated in the text above the table.

**Table 4 cancers-12-03756-t004:** Prediction based on HU and deviation

Groups Categorized by HU and Dev	Tumor Cells Present	Odds Ratio	Successful Molecular Analysis (≥30%)	Odds Ratio
Group 1 HU < 713.50 and Dev < 178.90	36/44 (81.8%)	A	34/44 (77.3%)	A
Group 2 HU ≥ 713.50 or Dev ≥ 178.90	10/18 (55.6%)	0.278 (*p* = 0.037)	8/18 (44.4%)	0.235 (*p* = 0.235)
Group 3 HU ≥ 713.50 and Dev ≥ 178.90	4/9 (44.4%)	0.178 (*p* = 0.026)	3/9 (33.3%)	0.147 (*p* = 0.016)

A: Group 1 was used as a reference to compare the other variables (group 2 and 3) to.

**Table 5 cancers-12-03756-t005:** Results of previous studies compared to the current study.

Reference	N	Imaging	Diagnostic Yield	Sufficiency for Molecular Analysis	Type of Molecular Analysis
[15]	80	CT-guided	69%	64%	RNA NGS ^‡^
[16]	115	Unguided	62.%	Not performed	
[9]	39	CT-guided	77%	Not performed	
[17]	43	MRI	72.1%	Not performed	
[18]	184	Unguided	25.5%	Not performed	
[19]	70	CT-guided	85.7%	WES ^†^ 81.7% ➔ RNA-seq 33.3%	DNA WES ^†^
[20]	54	CT-guided	67%	39%	RNA microarray analysis
[26]	10	CBCT- Guided *	90%	80%	Single molecular inversion probe and WES ^†^
Current study	110		76.4%	66.4% of total; 86.8% of biopsies with histological documentation of tumor cells	WES ^†^ and/or targeted NGS ^‡^ (possible when ≥30% tumor cells are available)

* Cone-beam CT-guided; ^†^ Whole-exome sequencing; ^‡^ Next-generation sequencing.

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
