# Peer review of "Prior PSMA PET-CT Imaging and Hounsfield Unit Impact on Tumor Yield and Success of Molecular Analyses from Bone Biopsies in Metastatic Prostate Cancer"

_cancers, 2020, doi:10.3390/cancers12123756_

Round 1
Reviewer 1 Report
Major comments:
Metastasis size is a missing data, not considered at all.
DNA concentration and quality should be reported and evaluate.
DNA extraction methods and type of molecular test performed are missing.
How the biopsy material has been processed? which fixative? has the material gone through decalcification process? molecular analysis was performed after histological evaluation? a fresh sample has been used for molecular analysis and another part was fixed in formalin?
Minor comments:
Please eliminate template stuff from your formatted manuscript
Biopsies from spine were more successful compared to other locations for both histological and molecular biopsies: replace biopsies with evaluation
Table 1. check rows alignment
wasseen: typos
Please replace molecular positive biopsy with "informative" or successful, positive is misleading
Author Response
Dear reviewer,
We would like to thank you for your time and constructive comments on our article ‘Prior PSMA PET/CT imaging and Hounsfield units impacts tumor yield and success of molecular analyses from bone biopsies in metastatic prostate cancer ‘. Please find below our responses to your comments, which have also been incorporated in our revised manuscript.
Major comments
Point 1: Metastasis size is a missing data, not considered at all.
Response 1: Metastasis size is indeed missing data as it is challenging to measure, thus not considered as an objective variable to be included. The ‘true’ size of a bone metastasis will influence the chances of success in sampling it. However, the established radiographical imaging modalities, such as the CT or bone scan, are not capable of robustly identifying tumor lesion size in the bone. On a CT scan, osteosclerotic lesions are commonly the result of either tumor-related sclerosis, or associate with a treatment effect, and not always with viable tumor. Bone scintigraphy indirectly measures sites of tumor metastases, but as it measures turnover of bone, size of a metastasis cannot be obtained. Also, on MRI a bone metastasis remains difficult to measure, and only a few MRI scans were included. We agree that PSMA-PET might more clearly define a ‘true’ metastasis size, based on tumor PSMA-avidic lesion. However, this will only be possible to measure in patients with prior PSMA-PET (in 62 of cases), and comparisons cannot be made with other imaging types.
As the Prostate Cancer Working Group has classified the size of osseous lesions as non-measurable by MRI, CT or bone scintigrapy, we have chosen to not consider this as a variable. Therefore, for this study we have not included tumor size, but we agree that for future studies, solely focusing on PSMA-PET-guided biopsies, tumor size should be included. We have commented this major comment in the discussion. size.
Point 2: DNA concentration and quality should be reported and evaluated
Response 2: We have now included an supplemental excel file where we included all DNA concentrations and tumor cell percentages of samples of our in-house biopsies as well as in the biopsies analyzed within the CPCT-02 trial. A major limitation is that all samples with low tumor content below 30% were not isolated, therefore DNA concentration were not assessed. DNA concentration of our in-house biopsies was assessed in biopsies with a tumor content ≥30%, assessed by an experienced pathologist. For NGS, varying input DNA was used (50 lower input and 200ng upper input), based on the eluate volume and concentration available. When evaluating all samples with successful NGS, no correlation between DNA concentration and tumor-cell percentage were found as we expected (r=0.196, p=0.115, n=66; nonparametric Spearman test), as even in the low tumor-cell percentages a non-tumoral (stromal) component is included. In the successful biopsies analyzed within the CPCT-02 trial (N=36) also no correlation was found between tumor content and DNA concentration (p=0.06).
As all samples were new tissue biopsies, and an experienced pathologist evaluated tissue quality as a QC check prior to DNA isolation, DNA quality was not routinely assessed in the workflow. This was only the case in damages tissue with e.g. crush artifacts. For our in-house NGS pipeline, DNA quality was very limitedly assessed. For the CPCT-02 trial, DNA quality was not assessed at all prior to WGS. To answer your query whether input DNA concentration was associated with low DNA quality, we could only query DNA quality in samples with sufficient stored DNA in our institution from our in-house biopsies.
To report on the quality of DNA, we assessed DIN scores in 10 successful in house biopsies and associated with DNA concentration and tumor content. No association was found between DNA concentration and DIN score (r=0.252, p=0.482, n=10) or between DIN score and tumor cell percentage (r=-0.197, p=0.584, n=10). Low quality was seen in both intermediate and low concentrations of DNA and high quality was seen in both intermediate and low concentrations of DNA. This was also the case for DNA quality and tumor content.
In conclusion we have included DNA concentration for samples where DNA was isolated. DNA quality could not be included for most patients, but we did not identify poor DNA quality as a major influencing factor in this study, and on the outcome of the success rate of bone biopsies.
Point 3: DNA extraction methods and type of molecular test performed are missing. How the biopsy material has been processed? which fixative? has the material gone through decalcification process? molecular analysis was performed after histological evaluation? a fresh sample has been used for molecular analysis and another part was fixed in formalin?
Response 3: We have now included these details on further processes of DNA extraction and molecular analyses within the CPCT-02 trial and of the in-house pipeline in the Method section. As extraction methods differed between CPCT-02 and in-house trial we show that this has not impacted results, as no differences were found in success rate between these methods. In short, DNA was isolated from fresh frozen tissue from the CPCT-02 trial by pulverization, and no decalcification was necessary for these biopsies. The biopsies analyzed in our in-house pipeline went through a decalcification process with EDTA. After that, DNA extracted and isolated as stated in the methods section.
Minor comments
Point 1: Please eliminate template stuff from your formatted manuscript
Response 1: The manuscript was written in the template file offered by the journal. We think that with making formatting changes by the editorial office of the journal to our original submitted manuscript, some errors of the template were included in this manuscript. We tried to eliminate these errors as much as possible in the revised manuscript, and have asked the editorial office of Cancers to help out.
Point 2: Biopsies from spine were more successful compared to other locations for both histological and molecular biopsies: replace biopsies with evaluation
Response 2: We changed this sentence into ‘Evaluation of spine biopsies showed more often successful results compared to other locations for both histological and molecular biopsies.
Point 3: Table 1. Check rows alignment
Response 3: In our originally submitted manuscript the rows alignment were correct in the tables 1, 2 and 3. We corrected these errors in the revised manuscript.
Point 4: wasseen
Response 4: Unfortunately this mistake was overseen in our submitted manuscript. We corrected the spelling in the revised manuscript
Point 5: Please replace molecular positive biopsy with "informative" or successful, positive is misleading
Response 5: We agree that ‘positive’ might be misleading and replaced the word ‘positive’ by ‘successful’ in the revised manuscript.
Reference: Trial Design and Objectives for Castration-Resistant Prostate Cancer: Updated Recommendations From the Prostate Cancer Clinical Trials Working Group 3. Scher et al, J Clin Oncol 2016
Sincerely,
Minke Smits
Reviewer 2 Report
The authors did a retrospective study to correlate the relationship between imaging modality and the tumor yield in biopsies in patients with metastatic prostate cancer. The size of sample is large enough to support their conclusion that prior PSMA-PET/CT imaging can improve biopsy outcome compared with CT and MRI. They also developed a predictive model to improve the success rate of bone biopsy by selecting the low HU and deviation region. The paper was well written and results are clearly shown in figures and tables. I recommend the acceptance of publication after minor revision.
- The simple summary is too long. Please follow the journal requirement to shorten it.
- I remember the journal of Cancers does not allow to provide sections of Background, Materials and methods, etc. Just state the whole story concisely.
- Figure 3. When you did the screen shot to get this figure, please make sure the red wave line is disabled for MMRd.
Author Response
Dear reviewer,
We would like to thank you for your time and constructive comments on our article ‘Prior PSMA PET/CT imaging and Hounsfield units impacts tumor yield and success of molecular analyses from bone biopsies in metastatic prostate cancer ‘. Please find below our responses to your comments, which have also been incorporated in our revised manuscript.
Point 1: The simple summary is too long. Please follow the journal requirement to shorten it.
Response 1: We agree that our original simple summary was too long and complicated and changed it to a shorter and more clear summary according to the author instructions by Cancers.
Point 2: I remember the journal of Cancers does not allow to provide sections of Background, Materials and methods, etc. Just state the whole story concisely
Response 2: It is indeed correct that the journal of Cancers allows abstracts as a whole story and without subheadings. We removed the headings and it is now written as a whole story in the revised manuscript.
Point 3: Figure 3. When you did the screen shot to get this figure, please make sure the red wave line is disabled for MMRd
Response 3: We had overseen this error in our manuscript, and we have now removed the red wave line in the revised manuscript
Sincerely,
Minke Smits
Round 2
Reviewer 1 Report
Thank you for endorse my comments into your revision.